# Peptide-guided functionalization and macrocyclization of bioactive peptidosulfonamides by Pd(II)-catalyzed late-stage C–H activation

Jian Tang[1], Hongfei Chen[1], Yadong He[1], Wangjian Sheng[1], Qingqing Bai[1] & Huan Wang [1]

Peptides and peptidomimetics are emerging as an important class of clinic therapeutics. Here we report a peptide-guided method for the functionalization and macrocyclization of bioactive peptidosulfonamides by Pd(II)-catalyzed late-stage C–H activation. In this protocol, peptides act as internal directing groups and enable site-selective olefination of benzylsulfonamides and cyclization of benzosulfonamides to yield benzosultam-peptidomimetics. Our results provide an unusual example of benzosulfonamide cyclization with olefins through a sequential C–H activation, which involves the generation of a reactive palladium-peptide complex. Furthermore, this protocol allows facile self-guided macrocyclization of sulfonamide-containing peptides by intramolecular olefination with acrylates and unactivated alkenes, affording bioactive peptidosulfonamide macrocycles of various sizes. Together, our results highlight the utility of peptides as internal directing groups in facilitating transition metal-catalyzed functionalization of peptidomimetics.

---

[1] State Key Laboratory of Coordination Chemistry, Jiangsu Key Laboratory of Advanced Organic Materials, School of Chemistry and Chemical Engineering, Nanjing University, Nanjing 210093, China. These authors contributed equally: Jian Tang, Hongfei Chen. Correspondence and requests for materials should be addressed to H.W. (email: wanghuan@nju.edu.cn)

 1

Peptides and peptidomimetics are emerging as clinic ther-aputics with high potency and selectivity.[1,2] One major driving force for the growing interest of these compounds is their capability in regulating protein–protein interactions, which have been identified in numerous disease-related biologica pro-cess. Therefore, it is highly desirable to develop chemical strate-gies to synthesize bioactive peptides and peptidomimetics with structural diversity. For example, arylsulfonamides and sultams (cyclic sulfonamide) are important pharmacophores in medicinal chemistry,[3–7] and introduction of sulfonamide functionality into peptides usually provides improved proteolytic stability, hydrogen-bonding possibilities and improved biological activ-ities.[8–11] Specifically, peptidomimetics containing benzofused sulfonamides or sultams are among the most potent inhibitors of disease-related proteases, and benzosultam derivatives often exhibit improved pharmaceutical properties (Fig. 1a).[12–15] Despite their promising bioactivities, the development of this class of compounds is hindered by the lack of facile synthetic methodologies, especially for the construction of benzosultam motifs. As an example, the 6,7-dichlorobenzothiazine unit of a calpain inhibitor (Fig. 1a) requires six steps of synthesis before conjugation to 2-amino-3-phenyl-propanal.[15] As direct con-struction of benzosultams by intermolecular cyclization is uncommon,[16–18] synthesis of benzosultams often relies on intramolecular cyclization of elaborated precursors, whose pre-paration is often challenging.[15,19–23] Despite recent advances, facile and efficient methods for the diversification and cyclization of peptidosulfonamides are still in demand.

Transition metal-catalyzed C–H activation has shown great promise for the functionalization of amino acids and late-stage modification of peptides, as demonstrated by Yu,[24–27] Lavilla/ Albericio,[28–30] Ackermann,[31–33] and Daugulis,[34] among oth-ers.[35–41] Peptides have been shown to coordinate with palladium through amide bonds, promoting the functionalization of neighboring C–H bonds. Herein, we report a peptide-guided method for the functionalization of peptidosulfonamides by Pd (II)-catalyzed late-stage C–H activation. This reaction has broad substrate scope and provides facile access to a variety of bioactive benzylsulfonamide- and benzosultam-peptidomimetics, as well as peptidosulfonamide macrocycles (Fig. 1b). This strategyutilizes the N-sulfonated peptides in substrates as internal direacting groups and requires no external ligand or removable directing group. In addition, the reactions to generate benzosultam motifs follow a Pd(II)-catalyzed sequential C(sp2)–H activation mechanism, in which a second Civation mechanism, in whin unusual reactive peptide–Pd(II) complex. Furthermore, this reaction protocol allows efficient macrocyclization of substrates bearing acrylates or unactivated alkenes to produce bioactive peptidosulfonamide macrocycles.

## Results

**Olefination of benzylsulfonamide peptide conjugates**. We initiated the investigation by evaluating the utility of dipeptide as a directing group to enable the olefination of benzylsulfonamides. To establish optimal reaction conditions, we employed an *ortho*-methylbenzylsulfonamide dipeptide conjugate **1a** and *tert*-butyl acrylate **2a** as substrates (Fig. 2). Detailed optimization studies reveal that the reaction proceeds most efficiently with 4.0 equiv of *tert*-butyl acrylate **2a** in the presence of 10 mol% Pd(OAc)$_2$ and 3.0 equiv of AgOAc in dichloroethane at 80 °C for 12 h, affording the *ortho*-olefination product **3aa** in 80% isolated yield (Supple-mentary Table 1). The exocyclic double bond in product **3aa** is determined to be *E*-configured by $^1$H-nuclear magnetic resonance ($^1$H-NMR) analysis (Supplementary Figure 1). It is noteworthy that in contrast with previous reports of Pd-mediated olefination

of arylsulfonamides,[42,43] no external ligand is required in this procedure. Replacement of the N-sulfonated dipeptide by $^t$Leu methyl ester (Supplementary Table 1, substrate **1a'**) completely abolished the reaction under standard conditions, indicating that the dipeptide is required to enable the olefination reaction.

Having established the optimal conditions, we sought to demonstrate the generality of this protocol with respect to the benzylsulfonyl group. As outlined in Fig. 2, substrates bearing *ortho*-halide-substitutions (**1b**, **1c**) are fully tolerated in this protocol by yielding the corresponding olefination products **3ba** and **3ca** in 78% and 82% isolate yields, respectively. Reaction of substrate **1d** gave dominantly the diolefination product **3da** with an overall 88% yield. Substrates with *para*-substitutions, including nitro-, chloro-, and methyl groups, all underwent facile olefination and afford the corresponding products in excellent yields (**3fa**-**3ga**). Substrates with *meta*-substitutions (**1h**-**1j**) reacted with *tert*-butyl acrylate **2a** smoothly; however, the yield of *m*-methylben-zylsulfonamide derivative **3j** was noticeably lower (63%) and the mono-olefination product appeared as the major product.

Next, we examined the scope of reaction with respect to alkenes. Using benzylsulfonamide dipeptide conjugate **1d** as the substrate, *t*-butyl, *n*-butyl, ethyl, and benzyl acrylate all react with substrate **1d** in high yields, indicating that the steric property of acrylate substitutions does not affect the reaction efficiency (**3db**-**3dd**). The olefination proceeded in good yields when vinyl ethyl sulfone, N, N-dimethylacrylamide and styrene were employed (**3de**–**3dg**), demon-strating the versatility of this reaction. Catalytic C–H olefination reactions with unactivated, aliphatic alkenes are generally challenging due to their intrinsic poor reactivity.[42–45] To our delight, unactivated alkenes 4-methylpent-1-ene and (allyloxy)benzene both reacted with high yields (**3dh**, **3bi**), further expanding the substrate scope of this chemistry. The utility of this protocol is further highlighted by labeling a fluorinated peptidosulfonamide **3b** with a fluorescent moiety in 70% isolated yield (**3bj**).

To extend the procedure to substrates with various peptide sequences, aliphatic amino acids, including Leu, Ile, and Val were placed at the N-terminus of the dipeptide. All substrates exhibited good reactivity and afforded diolefination products as the major products (Fig. 2, **3ka**–**3ma**). Altering dipeptide sequence to Gly-$^t$Leu in substrate **3n** lowered the reaction efficiency and lead to dominantly mono-olefination in 53% yield (**3na**). Moreover, we prepared benzylsulfonamide-tripeptide conjugates **3o** and **3p** with $^t$Leu-Gly-Val and $^t$Leu-Gly-Phe to challenge the versatility of this procedure with regard to the length of the peptides. We were pleased to find that these tripeptides were also efficient in promoting the *ortho*-C–H olefination, and products **3oa** and **3pa** were isolated in 63% and 67% yield, respectively. Together, these results demonstrate that peptides are powerful directing groups for the preparation of benzylsulfonamide peptidomimetics.

To address the potential epimerization issue during the reaction, substrates **3q** and **3q'** with dipeptides of D-$^t$leu-Ala and L-$^t$Leu-Ala were synthesized (Supplementary Fig. 2). We then conducted their reactions with acrylate **2a** under standard conditions, and the stereochemistry of the resulting products was evaluated by NMR and high performance liquid chromato-graphy (HPLC). Results showed that all substrates reacted efficiently with full conversion, affording corresponding products in high yields (Supplementary Fig. 2). Crude reaction mixtures of **3q** and **3q'** (both mono- and di-substituted products) gave distinct retention times when analyzed by reversed phase-HPLC, indicating that the stereochemical integrity was retained and no epimerization occurred under the reaction conditions.

**Cyclization of benzosulfonamide peptide conjugates**. To fur-ther explore the generality of this peptide-guided strategy, we

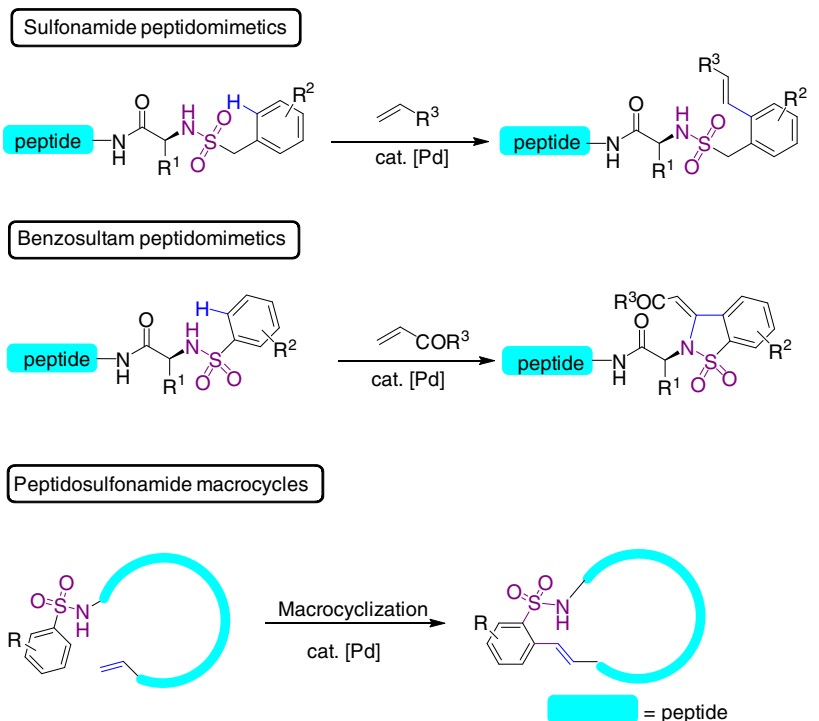

**Fig. 1** Synthesis of peptidomimetics containing aryl sulfonamide motif. **a** Bioactive benzylsulfonamide and benzosultam-containing peptidomimetics. **b** Peptide-guided functionalization and macrocyclization of sulfonamide-containing peptidomimetics by Pd(II)-catalyzed late-stage C–H activation. HAT human airway trypsin-like protease

next examine the reaction of benzosulfonamide peptide conjugates by employing a *p*-nitrobenzene-sulfonamide dipeptide conjugate **4a** and methyl acrylate **2k** as substrates (Fig. 3). Interestingly, instead of *ortho*-olefination, we found that a benzosultam-dipeptide conjugate **5ak** was generated as the major product, indicating that intermolecular cyclization has occurred. After extensively screening various parameters, we established that the treatment of substrate **4a** with 4.0 equiv of methyl acrylate **2k**, 12 mol% Pd(OAc)$_2$ as the catalyst, along with 2.0 equiv of CF$_3$COOAg, 2.0 equiv of Cu(OAc)$_2$, and 4.0 equiv of NaOAc in hexafluoroisopropanol at 80 °C for 24 h, affording the cyclization product **5ak** in 82% isolated yield (Supplementary Table 2). The requirement for a complex cocktail of metal reagents and base suggests that this reaction proceed through a distinct mechanism, which will be discussed in detail in the mechanism section.

With the optimized conditions in hand, we first examined the scope of this reaction with respect to the benzenesulfonyl group.

As outlined in Fig. 3, substrates bearing electron-withdrawing groups such as nitrile and trifluoromethyl, or electron-donating groups such as methyl, methoxyl, and phenyl all react efficiently to afford the benzosultam-dipeptide conjugates (**5ak–5gk**) in high yields. Halide-substituted substrates are fully tolerated (**5hk–5ik**, **5kk–5lk**), opening access to scaffolds that may be subject to further chemical manipulation. *Ortho-*, *para-*, and *meta-*substituents are all compatible with this procedure. Next, we evaluated the scope of acrylic acid esters (Fig. 3). Results show that ethyl acrylate, *n*-butyl acrylate, *t*-butyl, and benzyl acrylate all react with **4a** with high yields (**5aa–5ad**), indicating that the steric properties of acrylate substitutions do not affect the reaction efficiency. The cyclization proceeded in good yields when *N, N*-dimethylacrylamide and ethyl vinyl ketone (**5af–5ag**) were employed, further demonstrating the versatility of this method. However, no benzosultam products were obtained when styrene and 4-methylpent-1-ene was employed as substrates, indicating that a conjugated carbonyl group is important for cyclization.

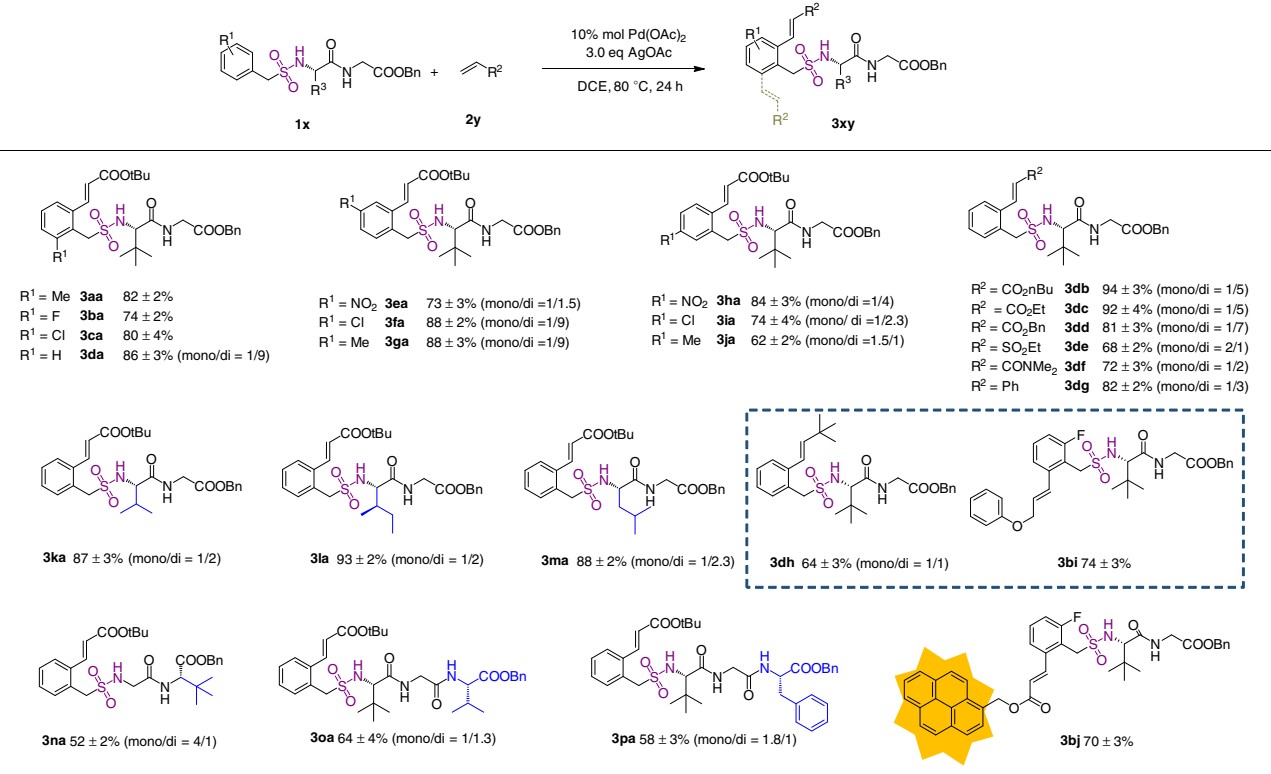

**Fig. 2** Scope of the olefination reaction of benzylsulfonamide peptide conjugates. The isolate yields were determined by three repeats of the experiments

To extend the procedure to other peptide conjugates, Leu, Ile, Val, and *t*-Boc-protected Thr were placed at the *N*-terminus of the dipeptide. Gratifyingly, all these substrates showed good reactivity and underwent cyclization affording the desired products (Fig. 3, **5ok–5rk**). For substrate **5s** containing a D-*t*leu, the benzosultam formation was equally efficient as substrate **5a**, indicating the chirality of amino acids in peptides has no impact on the reaction. Moreover, substrate **4t** with *t*Leu-Gly-Val tripeptide reacted with methyl acrylate efficiently, affording product **5tk** in 65% isolated yield. Together, these results demonstrate the versatility of this chemistry for the preparation of benzosultam peptidomimetics with peptides of various length and sequence.

To examine the potential epimerization during cyclization, substrates **4u** and **4v** with dipeptides of D-*t*leu-Ala and L-*t*Leu-Ala were synthesized and subject to reactions with methyl acrylate **2k** under standard conditions (Fig. 3). HPLC analysis showed that both reactions proceeded highly efficiently with full conversion, yielding single epimeric products **5uk** and **5vk** (Supplementary Fig. 3). Thus, our protocol is compatible with peptides without epimerization and the chirality of dipeptides has no impact on its efficiency as a directing group.

**Mechanism of benzosultam formation.** Previous reports of Pd- and Rh-catalyzed oxidative cyclization have generally proposed an aza-Wacker reaction following the initial olefination.[46–48] To gain insight into the mechanism of benzosultam cyclization in our protocol, we monitored the reaction progress of substrates **4m** and **2k** at various time points by liquid chromatography–mass spectrometry (Fig. 4). After 10 h reaction, the starting material **4m** was mostly consumed and converted to the corresponding olefination product **6mk** in 63% yield, and the cyclization product **5mk** in 19% yield. Prolonged reaction time

resulted in accumulation of the cyclization product **5mk**. Interestingly, an unexpected palladium-containing intermediate **7mk** (~10% yield) was observed during the course of reaction. Both NMR and X-ray crystallographic analysis revealed that **7mk** is a unique dipeptide–Pd(II) complex (Fig. 4b). In the structure of **7mk**, the slightly distorted square planar Pd(II) atom is bonded to the palladated alkene carbon and carbonyl oxygen of the newly installed methyl acrylate, and the sulfonamide nitrogen and amide oxygen of the dipeptide, by forming a tricyclic structure. Together, these results suggest that a second C–H activation of the olefination product **6mk** occurs and the dipeptide backbone directs the Pd(II) catalyst towards the proximal olefinic C(sp²)–H bond to generate the intermediate **7mk**. Treatment of purified compound **7mk** with 1, 2-bis(diphenylphosphino)-ethane (DPPE) (2.0 equiv) or reaction conditions omitting Pd catalyst resulted in near-quantitative conversion to the cyclization product **5mk**, indicating that **7mk** is a reactive intermediate for reductive elimination. In contrast, treatment of the olefination product **6mk** under reaction conditions in absence of Pd catalysis leads to no reaction, showing that Pd is required for benzosultam cyclization. Consistent with these observations, reactions of substrates **4a**, **4j**, and **4o** with methyl acrylate all generated the corresponding peptide–palladium complex **7ak**, **7jk**, and **7ok** as intermediates (Supplementary Fig. 4). Further treatment of these Pd complexes with DPPE yields the corresponding cyclized benzosultam peptidomimetics, thereby establishing the general nature of a second C–H activation.

The overall catalytic cycle is therefore proposed to proceed as follows (Fig. 4c). Following the insertion of Pd(II) into the *ortho*-C–H bond directed by the *N*-sulfonated peptide, the resulting intermediate **2** coordinates with an olefin **3**, and this is followed by 1,2-migratory insertion and β-hydride elimination, generating the olefination product **6**. Subsequently, this uncyclized intermediate **6** associates with the Pd(II) catalyst again and undergoes

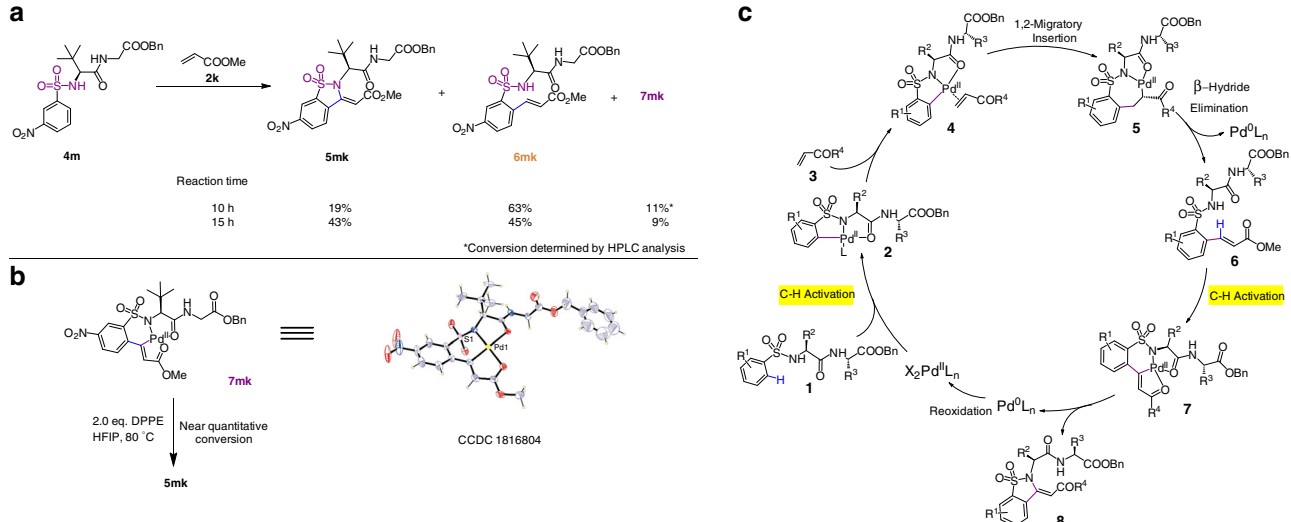

**Fig. 3** Scope of the cyclization reaction of benzosulfonamide peptide conjugates. The isolate yields were determined by three repeats of experiments

**Fig. 4** Mechanistic investigation of the benzosultam cyclization reaction. **a** Monitoring of the reaction progress at various time points by LC–MS. **b** X-ray crystallographic analysis of the Pd(II)–peptide complex **7mk** and its conversion to product 5mk in the presence of DPPE. ORTEP representation of the crystal structure of **7mk**. Gray = carbon, blue = nitrogen, purple = sulfur, yellow = Pd. Condition: 20 mol% Pd(OAc)₂, 2.0 eq. AgTFA, 2.0 eq. Cu(OAc)₂, 4.0 eq. AgOAc, HFIP, 80 °C. **c** Proposed catalytic cycle for olefination/oxidative cyclization

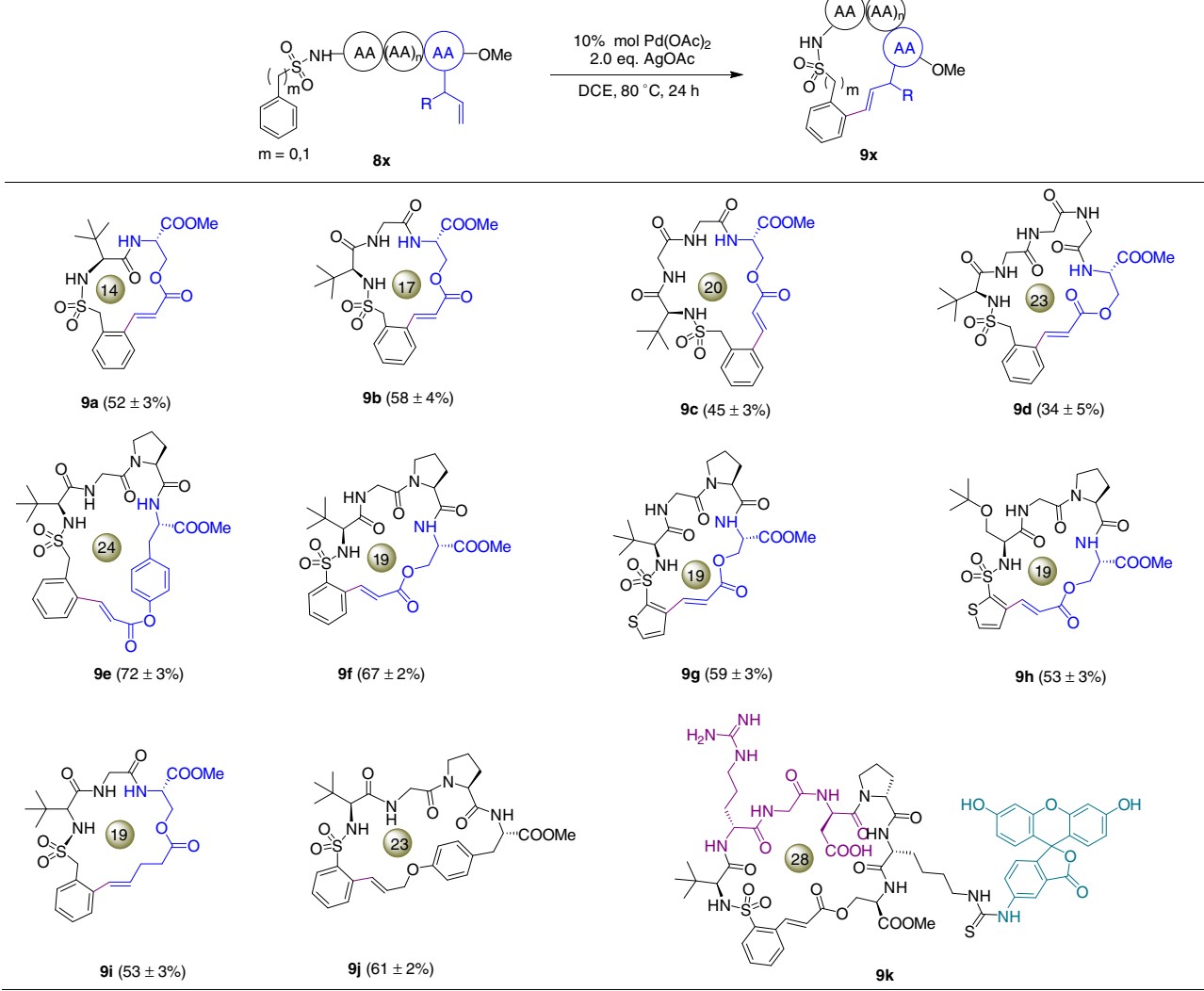

**Fig. 5** Macrocyclization of peptidosulfonamides by self-guided C–H olefination. **a** Synthesis of peptidosulfonamide macrocycles. **b** FITC-labeled cyclic RGD peptide **9k** and its integrin binding assays analyzed by confocal microscopy. All peptides were used at 2 μM concentration for the cell binding studies. The RGD motif is in red and the fluorescein motif is in green. The isolate yields were determined by three repeats of the experiments

a second C–H activation at the newly installed olefin by forming a tricyclic intermediate **7**. Finally, reductive elimination of this Pd (II) complex intermediate **7** yields the benzosultam-dipeptide conjugate **8** as the final product. Reoxidation of $Pd^0L_n$ by Ag(I) or Cu(II) generates $X_2Pd^{II}L_n$ catalyst to close the catalytic cycle. Our results provide an unusual example of cyclization of benzosulfonamide derivatives with olefins through a sequential C–H activation mechanism.

**Self-guided macrocyclization of peptidosulfonamides.** Cyclic peptides and peptide-based macrocycles have shown broad structural diversity and biological functions, however, chemical methods of constructing peptide macrocycles are currently limited.[49-51] To further explore the applicability of our method, we conducted peptide macrocyclization to generate bioactive peptidosulfonamide macrocycles. We started with a benzylsulfonamide dipeptide conjugate **8a** with the sequence of $^t$Leu-O-acryloylserine-OMe (Fig. 5a). Gratifyingly, substrate **8a** underwent efficient macrocyclization through o-olefination under standard conditions, affording the 14-membereddipeptide macrocycle **9a** in 53% yield. Encouraged by this result, we continued to examine the macrocyclization of tripeptide, tetrapeptide and pentapeptide substrates **8b–8e**. All

substrates cyclized efficiently and provided 17- to 24-membered macrocycles **9b–9e** in good yields, demonstrating the versatility of our strategy in constructing peptide macrocycles. Similarly, benzosulfonamide tetrapeptide conjugate **8f** also cyclized under standard conditions, yielding a 20-membered macrocycle **9f**.

We next challenged our method by introducing heterocycles into cyclic peptides. Thiophene is an important building block in a number of drugs and natural products.[52] Thiophene-sulfonamide peptide conjugates **8g** and **8h** were then synthesized and subject to the standard procedure, and to our delight, macrocyclization was achieved in high yields through *ortho*-olefination of thiophene motifs, resulting in two 19-membered macrocycles **9g** and **9h**. The exocyclic double bond in product **9h** is determined to be *E*-configured by [1]H-NMR analysis (Supplementary Fig. 1). Olefination of benzylsulfonamides with unactivated alkenes is generally challenging and has not been reported in macrocyclization of peptides.[53] Following our method, substrate **8i** and **8j** bearing unactivated alkene groups also underwent macrocyclization smoothly, further expanding the substrate scope of this protocol (Fig. 5). Together, these results indicate that the N-sulfonated peptide is powerful in self-guiding intramolecular C–H olefination. Following this self-guided

peptide macrocyclization method, we proceeded to synthesize a fluorescent-labeled cyclic RGD peptide **9k** with fluorescein isothiocyanate conjugated to a Lys residue (Supplementary Fig. 5). The RGD sequence is known for its selective binding to integrins when incorporated in appropriate cyclic structures.[54] We examined the binding of compound **9k** to U87MG cells, which is a glioblastoma cell line overexpressing the αvβ3 integrin. U87MG cells were then incubated with cyclic peptide **9k** for 90 min before subjected to confocal microscopy analysis (Supplementary Fig. 6). Strong fluorescence staining was observed, indicating that the peptidosulfonamide macrocycle **9k** exhibited binding affinity to the integrin. Together, these results demonstrate the applicability of our method in generating bioactive peptidosulfonamide macrocycles as molecular tools in biological systems.

## Discussion

In summary, we have developed a peptide-guided method for functionalization and macrocyclization of bioactive peptidosulfonamides by Pd-catalyzed C–H activation. The potency of the peptides as a directing element is well demonstrated by the site-selective olefination of benzylsulfonamide with both activated and unactivated alkenes, as well as the cyclization of benzosulfonamide with acrylates. Moreover, our protocol allows macrocyclization of peptidosulfonamides bearing alkene functionalities. A RGD-containing peptidosulfonamide macrocycle exhibits strong binding affinity towards integrins, demonstrating the potential application of this chemistry in the development of pharmaceutical compounds and biological probes. The increasing significance of bioactive peptidomimetics containing pharmacophores should render this method attractive for synthetic and medicinal chemistry.

## Methods

**General information**. All the reagents and solvents were obtained from Sigma-Aldrich, Alfa-Aesar or Acros, and used directly without further purification. Amino acids and derivatives were obtained from commercial sources. NMR spectra were recorded on Bruker AMX-400 instrument for $^1$H-NMR at 400 MHz and $^{13}$C NMR at 100 MHz, using TMS as internal standard. The following abbreviations (or combinations thereof) were used to explain multiplicities: s = singlet, d = doublet, t = triplet, q = quartet, m = multiplet, br = broad. Coupling constants J, are reported in Hertz units (Hz). High-resolution mass spectra (HRMS) were recorded on an Agilent Mass spectrometer using electrospray ionization-time of flight. HPLC profiles were obtained on Agilent 1260 HPLC system using commercially available columns.

**General procedure for Pd-catalyzed reactions**. Typically, the peptidosulfonamide substrates were placed in a 15 ml sealed reaction tube under indicated reaction conditions. The reaction mixture was stirred at 80 °C for 12–24 h, cooled to room temperature and then diluted with EtOAc (5.0 ml). The resulting solution was filtered through a Celite pad, concentrated under reduced pressure and the product was further purified by column chromatography and was typically obtained as a white solid.

**Cell culture and staining experiments**. U87MG cells were grown and maintained in DMEM media with 10% FBS and 1% penicillin/streptomycin at 37 °C, 5% CO$_2$. Before staining experiments with peptides, the cells were seeded on the surface of MatTek glass bottom microwell dishes using 1 mL media. After 1 day, the cells were washed twice with warm DMEM media, incubated at 37 °C with 2 µM peptide **9k** for 90 min and fixed. Images were taken using a Leica TCS SP8 confocal fluorescence microscope.

**Data availability**. X-ray crystallographic data of compound **7mk** has been deposited in The Cambridge Crystallographic Data Centre (CCDC) with the Accession number of 1816804. All relevant data are available in supplementary information and from the authors.

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

## Acknowledgments

This work is supported by the 1000-Youth Talents Plan, NSF of China (Grant 21778030), NSF of Jiangsu Province (Grant BK20160640) and the Fundamental Research Funds for the Central Universities (Grants 14380138 and 14380131). We thank Prof. Shaolin Zhu (NJU) and Prof. Zhuangzhi Shi (NJU) for helpful discussion during the preparation of the manuscript.

## Author contributions

T.J. and H.C. designed and carried out most of the chemical reactions and analyzed the data. Y.H. and Q.B. supported the design and performance of synthetic experiments. W.S. performed the cell culture and staining experiments. H.W. designed and supervised the project. All authors discussed the results and commented on the manuscript. T.J., H.C., and H.W. wrote the manuscript. All authors have given approval to the final version of the manuscript.

## Additional information

**Competing interests:** The authors declare no competing interests.

