## [Peer Review File · Nature Communications]

Reviewer #1 (Remarks to the Author):

The paper by Wang and coworkers describes a late-stage peptide-guided, Pd-catalyzed C-H functionalization. The work is interesting and will have sufficient impact for publishing in Nature Communications after the following minor revisions.

(1) Authors use enantiopure starting materials. For a few (2 or so) products in Schemes 1-2 please determine if any racemization has occurred during the reaction.

(2) Please explain in SI how stereochemistry around exocyclic double bond was determined.

(3) A few typos need to be fixed: page 3, last line - diphenyl, in Abstract - post-translational modified.

(4) While the proposed mechanism makes sense, please submit 5ma or a similar compound to reaction conditions in absence of Pd catalyst.

(5) SI is very incomplete and sketchy. Please provide exact chromatography solvents/R_f values for every compound that was chromatographed, retention times for compounds purified by HPLC, etc.

(6) Ref 2 is extremely strange - what format does it follow?

Reviewer #2 (Remarks to the Author):

(a). The title does not much sense (scientifically) and the analogy re leader sequences is at best flimsy and actually (I believe) erroneous as the catalyst in reality binds to the pharmacophore (the blue box) – not remotely as in a leader peptide. Indeed the Pd binding site is modified and becomes part of the cyclised product. Figure 1 is thus a little misleading re the chemistry that takes place and all the arguments of biological analogy with leader sequences and post-translational modification is wrong.

(b). Remote functionalization is well known in a huge number of areas e.g. the work of Barton and steroid functionalization and also see the work of Breslow e.g. see Biomimetic control of chemical selectivity. *Acc. Chem. Res.* 13, 170–177 (1980) – and “Simple amine-directed meta-selective C–H arylation via Pd/norbornene catalysis. *J. Am. Chem. Soc.* 137, 5887–5890 (2015). What is shown here is just another variant of this. There are many reports of remote functionalization/directing groups in Pd chemistry.

(c). The larger ring based cyclisation is interesting – but it is a single example and again is NOT as shown re figure 1.

(d). Figure 2a – are these drugs? Where is the stereochemistry in the Calpain inhibitor?

(e). What is an oligopeptide?

(f). The cell based work is weak, it shows poor quality images and it adds little value.

Reviewer #3 (Remarks to the Author):

Wang and coworkers reported a new palladium catalyzed ortho C-H functionalization reaction of sulfonamides of short peptides with acrylate to synthesize cyclic benzosulfamides (benzosultams). In addition, they demonstrated this ortho C-H functionalization can proceed in an intramolecular manner to generate sulfonamide linked peptide macrocycles. Pd-catalyzed ortho C-H olefination of sulfonamides with acrylate has been previously reported. In this work, the authors discovered that the ortho C-H olefination intermediate of benzosulfamides of dipeptide and tripeptide can undergo another step of C-H functionalization to give a cyclized benzosultam product. Similar double C-H action reactions of arenes with acrylates are known. These cyclization reactions proceed in good yield on simple substrates. Substituents on sulfonamides are well tolerated. The proposed reaction mechanism is reasonable. However, there are several issues with this chemistry: 1) Very high loading of catalyst (20 mol % of Pd(OAc)₂) and a complex cocktail of metal reagents/base (2 equiv of AgTFA, 2 equiv of Cu(OAc)₂, and 4 equiv of NaOAc) at relatively high reaction temperature are required. The complexity and relatively harsh reaction conditions would limit the applicability of this reaction to complex peptide substrates. 2) Only acrylates and similar olefin analogs were tested. 3) Only one example of macrocyclization of a special peptide substrate was demonstrated (no details on the synthesis of compound 3va-FITC are provided).

While this work demonstrates an interesting application of Pd-catalyzed C-H functionalization of peptide substrates, it appears that considerable improvement is still needed for publication on journal like Nature Communications. The authors could consider submitting the current form of this work to a more specialized chemistry journal such as European Journal of Chemistry.

Response to Referees Letter

To address the comments and concerns from the reviewers, we have made substantial revision to the original manuscript, including:

1) The analogy of our chemistry to the enzymatic modifications during RiPPs biosynthesis (Fig. 1A) might be confusing with the concept of “remote functionalization” (Reviewer #2). Therefore, we have provided an updated Fig. 1 for a better demonstration, as well as a detailed explanation in the response to Reviewer #2.

2) To demonstrate the applicability of our methodology (Reviewer #3), we extend the substrate scope to benzy sulfonamides (**24 additional examples**), demonstrating that the oligopeptides can facilitate the olefination of benzy sulfonamides with remarkable efficiency. **The reaction condition is mild, and importantly, both acrylates and unactivated alkenes are accepted as substrates.**

3) Regarding the harshness of reaction conditions for benzosultam cyclization (Reviewer #3), we performed further optimization. Gratifyingly, we were able to lower the loading of Pd(OAc)₂ from 20% to 12 mol%, temperature from 100 °C to 80 °C, without affecting the reaction yields.

4) Both reviewer #2 and #3 thought the macrocyclization of peptidosulfonamides is interesting and required more examples. Therefore we provide **10 additional examples** of cyclic peptides with ring sizes ranging from 14- to 28-membered macrocycles. Importantly, we demonstrated that heterocycles, such as thiophene, and unactivated alkenes can be incorporated into macrocycles using our methods.

In addition to these major revisions, we have carefully addressed all the comments from the reviewers, provided missing/additional data to better support our conclusions and revised the manuscript accordingly. A

point-by-point response to reviewers' comments was attached at the end of this letter and the details of response are listed below.

Point-to-point response:

In response to Reviewer #1 (quotes from reviewer are in italicized):

The paper by Wang and coworkers describes a late-stage peptide-guided, Pd-catalyzed C-H functionalization. The work is interesting and will have sufficient impact for publishing in Nature Communications after the following minor revisions.

(1) Authors use enantiopure starting materials. For a few (2 or so) products in Schemes 1-2 please determine if any racemization has occurred during the reaction.

Response to the comment:

Thanks for the comment.

To address the potential epimerization issue during the reaction, we synthesized substrates **3q** and **3q'** with dipeptide auxiliaries of *D*-tLeu-Ala and *L*-tLeu-Ala (Page 3, Paragraph 3; Scheme S2). Two substrates reacted with acrylate **2a** under standard conditions, and the stereochemistry of the resulting products was evaluated by NMR and RP-HPLC. Results showed that all substrates reacted efficiently with full conversion, affording corresponding products in high yields. Both reactions yielded single epimeric products with distinct retention times when analyzed by RP-HPLC, indicating that the stereochemical integrity was retained and no epimerization occurred under the reaction conditions.

Similarly, benzosulfonamide substrates **4u** and **4v** with dipeptide auxiliaries of *D*-tLeu-Ala and *L*-tLeu-Ala were synthesized and subjected to reaction

conditions (Scheme S3). RP-HPLC analysis showed that single epimeric product was generated for each substrate, indicating that no epimerization occurred during the cyclization of benzosultams.

(2) Please explain in SI how stereochemistry around exocyclic double bond was determined.

Response to the comment:

To address this question, we have provided ¹H-NMR of product **3aa** and cyclic peptide **9h** in Scheme S1. The coupling constant of the resulting exocyclic double bonds was determined to be $J_{ab}=15.0$ Hz, and therefore we conclude that the double bonds are in *E*-configuration.

(3) A few typos need to be fixed: page 3, last line - dipnyl, in Abstract - post-translational modified.

Response to the comment:

The typos have been corrected.

(4) While the proposed mechanism makes sense, please submit 5ma or a similar compound to reaction conditions in absence of Pd catalyst.

Response to the comment:

As requested, compound **7mk** (compound **5ma** in the original manuscript, Scheme 3A) was prepared and subject to the reaction conditions without Pd catalyst (2.0 equiv of CF₃COOAg, 2.0 equiv of Cu(OAc)₂, and 4.0 equiv of NaOAc in HFIP at 80 °C). Results showed that compound **7mk** was converted to the corresponding cyclization product **5mk**, but in a lower yield compared to the reaction with DPPE treatment. This result supports our mechanistic proposal that the formation of C-N bond indeed follows a sequential C-H activation mechanism as proposed in Scheme 3C.

(5) SI is very incomplete and sketchy. Please provide exact chromatography solvents/R_f values for every compound that was chromatographed, retention times for compounds purified by HPLC, etc.

Response to the comment:

Thanks for the comment.

We have revised the supporting information and provided detailed information on purification (such as exact chromatography solvents/R_f values) for each compound.

Please see “SI-Secton 4: Spectra data” for details.

(6) Ref 2 is extremely strange - what format does it follow?

Response to the comment:

The format of reference 2 (Ref 4 in the revised manuscript) has been corrected.

In response to Reviewer #2

(1). The title does not much sense (scientifically) and the analogy re leader sequences is at best flimsy and actually (I believe) erroneous as the catalyst in reality binds to the pharmacophore (the blue box) – not remotely as in a leader peptide. Indeed the Pd binding site is modified and becomes part of the cyclised product. Figure 1 is thus a little misleading re the chemistry that takes place and all the arguments of biological analogy with leader sequences and post-translational modification is wrong.

Response to the comment:

Thanks for the comment.

As our original manuscript might be unclear regarding the analogy of our chemistry to the leader peptide-guided enzyme modification, we provide an updated Fig. 1 for better illustration. Although it is not a “perfect” analogy, in the following explanation, we hope to convince the reviewer that it is a reasonable analogy.

Explanation:

As an analogy between an enzyme (biocatalyst) and a transition metal catalyst in general, the active site of an enzyme is equivalent to the reactive metal center, which is the key component that participates in chemical transformations. The bulky portion of the enzyme surrounding the active site is equivalent to the ligands of a metal catalyst, creating a microenvironment to facilitate and tune the metal center to catalyze reactions. For an enzyme to function, the active site needs to bind/associate **directly** to the target amino acid residue for modification, just as the Pd catalyst binds directly to the pharmacophore.

During the biosynthesis of RiPPs, leader binding and active site-mediated modification by enzymes are usually two separate processes. Enzymes in RiPPs biosynthesis typically have two distinct sites: one is the leader binding site, which is designed to bind the leader portion of the peptide substrates; the other is the catalytic active site to execute chemical transformation. The modification enzyme will first bind to the leader peptide, and then the active site will bind to the residue to be modified. (First paragraph, Fig. 1A). The original Figure 1A might be somehow misleading and give the reviewers the impression of “remote functionalization”, therefore we have updated it for better demonstration.

In the reactions of benzylsulfonamide-peptide conjugates (Scheme 1), the oligopeptide auxiliary (like the leader peptide) associates with Pd catalyst (like the enzyme) first, placing the Pd center (like the active site of the enzyme) in an ideal position to bind and activate the benzylsulfonamide motif through

the formation of six-member-ring cyclopalladation. The Pd binding-site is not modified in this case. The reactions of benzosulfonamide-peptide conjugates (Scheme 2) follows the same logic, only in that case, the nitrogen of sulfonamide group is involved in cyclization during the sequence C-H activation reaction.

Since our manuscript is trying to demonstrate the potency of oligopeptides as internal directing auxiliary in complex peptidomimetics, we think, at least in part, that our chemistry is in good analogy to the enzyme modification in *RiPPs* biosynthesis.

(2). Remote functionalization is well known in a huge number of areas e.g. the work of Barton and steroid functionalization and also see the work of Breslow e.g. see Biomimetic control of chemical selectivity. Acc. Chem. Res. 13, 170–177 (1980) – and “Simple amine-directed meta-selective C–H arylation via Pd/norbornene catalysis. J. Am. Chem. Soc. 137, 5887–5890 (2015). What is show here is just another variant of this. There are many reports of remote functionalization/ directing groups in Pd chemistry.

Response to the comment:

Thanks for the comment and recommended literature. Prof. Breslow has always been an inspiration, and Prof. Dong is a leading chemist in transition metal catalysis and we have learned a great deal from his work.

As partially explained in **Comment #1**, instead of “remote functionalization”, this manuscript is to demonstrate the potency of oligopeptides in guiding metal catalysts for site-selective functionalization of peptidomimetics. Although directing groups have been extensively employed in a huge number of studies, the potential of oligopeptide backbones as directing components in complex molecules (such as cyclic peptides) has not been well explored. As the importance of peptidomimetics and peptide-drug conjugates

(similar to antibody-drugs conjugates) raises over the years, we hope to provide a methodology for the synthesis and facile functionalization of this type of compounds. Furthermore, it might promote the application of “transition-metal catalyzed C-H activation” in site-selective protein modifications, as Davis et al have demonstrated in one example that amino acid sidechains with metal binding capacity can provide site-selectivity to Pd-catalyzed cysteine arylation of proteins (J. Am. Chem. Soc. 2016, 138, 8678–8681).

Our method requires no external ligand or designated directing groups that require additional removal steps. In addition, our work shows that the participation of oligopeptides in catalysis results in interesting chemistry, providing the first example of cyclization of benzosulfonamides with olefins through a sequential C–H activation, as previous studies generally propose an aza-Wacker mechanism. (Scheme 3).

We agree with the reviewer that our work utilizes directed Pd-catalyzed C-H functionalization. At the meantime, we believe that our work provide important knowledge regarding the chemistry and application of oligopeptides in the functionalization of peptide-pharmacophore conjugates, as well as the synthesis of peptidomimetic macrocycles.

(3). The larger ring based cyclisation is interesting – but it is a single example and again is NOT as shown re figure 1.

Response to the comment:

To further demonstrate the applicability of our method in peptide macrocyclization, we provide 10 additional examples (compounds **9a-9k**) of macrocycles (Scheme 4). The sizes of the resulting macrocycles range from 14- to 28-membered rings with various sequences. Importantly, we demonstrated that heterocycles, such as thiophene, and unactivated alkenes can be incorporated into macrocycles using our methods.

(4). Figure 2a – are these drugs? Where is the stereochemistry in the Calpain inhibitor?

Response to the comment:

Thanks for the comment.

Compounds in Fig. 2A are not clinical drugs, but have shown promising potential in the treatment of related diseases. We have changed the subtitle of Fig. 2A to “*Bioactive sulfonamide/benzosultam-containing peptidomimetics*” for a more accurate description.

The stereochemistry of calpain inhibitor was indeed missing in the original manuscript and has been corrected.

(5). What is an oligopeptide?

Response to the comment:

An “Oligopeptide” typically refers to a “short peptide”, which consists of 2 to 20 amino acids. Similarly, a “polypeptide” usually refers to a long and unbranched peptide shorter than 50 amino acids. However, there is no strict definition about the length of oligopeptides and polypeptides, and sometimes they are just called “peptides”. In our manuscript, peptide auxiliaries are typically 2-7 amino acids, and therefore are referred as “oligopeptides”.

(6). The cell based work is weak, it shows poor quality images and it adds little value.

Response to the comment:

Thanks for the comment.

The cell-based assay in the manuscript demonstrated the bioactivity of macrocycle **9k** containing a RGD motif, which is an important bioactive peptide sequence extensively used in chemical biology and medicinal chemistry. The cell staining assay is a typical assay to examine the uptake and bioactivity of

cyclic peptides. Similar demonstration can be found in *Nat. Comm.* 2015, 6, 7160. (Fig. 4C), *J. Am. Chem. Soc.* 2016, 138, 2098–2101.(Fig. 4) and *Chem. Sci.*,2017, 8,4565–4570 (Fig. 4).

To provide a result with improved image quality, we have performed the cell staining assay again. Please see Scheme S6.

Reviewer #3 (Remarks to the Author):

Wang and coworkers reported a new palladium catalyzed ortho C-H functionalization reaction of sulfonamides of short peptides with acrylate to synthesize cyclic benzosulfamides (benzosultams). In addition, they demonstrated this ortho C-H functionalization can proceed in an intramolecular manner to generate sulfonamide linked peptide macrocycles. Pd-catalyzed ortho C-H olefination of sulfonamides with acrylate has been previously reported. In this work, the authors discovered that the ortho C-H olefination intermediate of benzosulfamides of dipeptide and tripeptide can undergo another step of C-H functionalization to give a cyclized benzosultam product. Similar double C-H action reactions of arenes with acrylates are known. These cyclization reactions proceed in good yield on simple substrates. Substituents on sulfonamides are well tolerated. The proposed reaction mechanism is reasonable. However, there are several issues with this chemistry:

1) Very high loading of catalyst (20 mol % of Pd(OAc)₂) and a complex cocktail of metal reagents/base (2 equiv of AgTFA, 2 equiv of Cu(OAc)₂, and 4 equiv of NaOAc) at relatively high reaction temperature are required. The complexity and relatively harsh reaction conditions would limit the applicability of this reaction to complex peptide substrates.

Response to the comment:

Thanks for the comment.

In order to address the concern regarding to the harshness of reactions conditions and substrate tolerance, we have provided additional data as following:

1. We expand the substrate scope to benzylsulfonamide oligopeptide conjugates (Scheme 1). We show that peptide-guided *ortho*-olefination of benzylsulfonamides occurs under relatively mild conditions: 10 mol% Pd(OAc)₂ and 3.0 equiv of AgOAc at 80 °C for 20 h. Importantly, both acrylates and unactivated alkenes are good substrates for this reaction, which addresses the reviewer's second concern.
2. For the reaction of benzosulfonamide oligopeptide conjugates (Scheme 2), we further optimized the reaction conditions to: 12 mol% Pd(OAc)₂, 80 °C, 24 h. During revision, we used a brand-new temperature controlling module, and better temperature control is believed to be the major factor for the improvement.
3. The cocktail of metal reagents are still required. Omission of Cu(OAc)₂ from the reaction results in accumulation of the Pd-oligopeptide complex (such as **7mi**, Scheme 3) and lower yields of benzosultam cyclization products. In addition, we need to point out that only acrylates and substrates that contain an adjacent carbonyl group can lead to benzosultam formation, which is required by the reaction mechanism.
4. We have provided 10 additional examples of cyclic peptides with 14- to 28-membered rings, demonstrating the applicability of this reaction to complex peptide substrates.

2) Only acrylates and similar olefin analogs were tested.

Response to the comment:

As demonstrated by substrates **3dh**, **3bi** (Scheme 1) and cyclic peptides **9i**, **9j** (Scheme 3), unactivated olefins are also good substrates in our protocol.

3) Only one example of macrocyclization of a special peptide substrate was demonstrated (no details on the synthesis of compound 3va-FITC are provided).

Response to the comment:

To further demonstrate the applicability of our method in peptide macrocyclization, we provide 10 additional examples (compounds **9a-9k**) of macrocycles. The sizes of macrocycles range from 14- to 28-membered with various sequences. Both acrylates and unactivated alkenes can be used for macrocyclization.

Reviewer #1 (Remarks to the Author):

Can be published as-is now.

Reviewer #2 (Remarks to the Author):

I think this is much improved indeed. The title is now strong and the chemistry is a valuable addition to the literature.

I still think that the analogy re post-translational modifications and leader sequences is weak (at best) and irrelevant. It actually detracts from the key messages of the paper – thus the introduction has nothing to do with the teachings of the paper. There needs to be some thought about who the audience is for this paper – If I was a synthetic chemist (to whom this paper is really targeted) I would have stopped reading after the first line.... “post-translational modification...”.

The leader peptide analogy also does not work as the authors have show that it can be modified/alterd without affecting its function. It is also fully incorporated into a cyclic sulphonamide/peptide. I strongly recommend that the authors just remove it and use the liberated space in the introduction to cover more appropriate elements.

The term auxiliary is used in the paper – is this correct as it becomes part of the product ?

I am a peptide chemist. The term oligopeptide is a tautology and is wrong. Oligo means many units, while a peptide is (by definition) multiple amino acids joined together – the term that should be used is peptide – NOT oligopeptide (as this formally means many peptides joined together).

The sentence suggesting compatibility with solid-phase synthesis just because a Boc group remained intact is pushing credibility. This needs to either proven (by doing the chemistry on a resin) or the claim removed. The use of Pd and other salts is a real issue for SPPS (often blocks the solid-phase), while some linkers are cleaved by cold HFIP. HFIP at 80oC with AgTFA – who knows.

By definition (IUPAC) the abbreviation for d-amino acids is a three letter code with no capitals, while the L-amino acids has the first letter capitalised. tLeu needs to be rewritten - italics, superscript.

I would like to see yields given with an n=3 i.e. three repeats and +- SD to show robustness of the chemistry. This should be compulsory.

Reviewer #3 (Remarks to the Author):

In this revised manuscript, Wang and coworkers slightly improved the reaction conditions and expanded the substrate scope of this palladium catalyzed sulfonamide directed ortho C-H olefination with alkenes. Compared with the previous version using benzenesulfonamide substrates, benzylsulfonamides with additional methylene group were able to undergo the same ortho C-H olefination reaction. This type of Pd catalyzed ortho aryl C-H functionalization particularly with activated olefins have been widely reported in recent literature with all kinds of directing groups. Their applications with peptide substrates for structural modification and cyclization are also becoming common. The first part of this work, the intermolecular C-H olefination, is quite routine for directed C-H olefination. The second part, macrocyclization, is relatively more interesting. However, it is more of a proof of concept using simple protected peptide substrates. Most of the peptide substrates are very similar. It is fair to say that it offered no advantage compared with the conventional macrolactonization or macrolactamization chemistry, which can easily access the products generated via this sophisticated Pd-catalyzed method. This chemistry would have appeared more interesting if discovered three years ago. Overall, this work shows a new application of method-catalyzed C-H functionalization chemistry on peptide substrates, however, it lacks the level of novelty and significance to publish in Nature Communications. The authors could consider submitting the current form of this work to a more specialized chemistry journal such as European Journal of Chemistry.

Nanjing University
School of Chemistry and
Chemical Engineering
Nanjing, 210093, China

Huan Wang
Professor of Chemistry
Room E504, Chemistry Building
Phone: (86-25) 8968-2133
E-mail: wanghuan@nju.edu.cn

May 12th, 2018

Dear Dr. Bottari,

Regarding our manuscript titled "Peptide-Guided Functionalization and Macrocyclization of Bioactive Peptidosulfonamides by Pd(II)-catalyzed Late-stage Activation" (NCOMMS-18-04330), we are providing our response to the **reviewer #2** in a point-by-point manner. Quotes from the reviewer are in italicized and our responses are marked in yellow.

Reviewer #2 (Remarks to the Author):

I think this is much improved indeed. The title is now strong and the chemistry is a valuable addition to the literature.

I still think that the analogy re post-translational modifications and leader sequences is weak (at best) and irrelevant. It actually detracts from the key messages of the paper – thus the introduction has nothing to do with the teachings of the paper. There needs to be some thought about who the audience is for this paper – If I was a synthetic chemist (to whom this paper is really targeted) I would have stopped reading after the first line.... “post-translational modification...”.

The leader peptide analogy also does not work as the authors have show that it can be modified/altered without affecting its function. It is also fully incorporated into a cyclic sulphonamide/peptide. I strongly recommend that the authors just remove it and use the liberated space in the introduction to cover more appropriate elements.

Response:

We agree with the reviewer that the leader sequence analog may be distracting and therefore remove it from the introduction section.

The term auxiliary is used in the paper – is this correct as it becomes part of the product ?

Response:

We agree with the reviewer that the term auxiliary is not accurate and have changed them to terms such as “directing group”. Please see the revised manuscript.

I am a peptide chemist. The term oligopeptide is a tautology and is wrong. Oligo means many units, while a peptide is (by definition) multiple amino acids joined together – the term that should be used is peptide – NOT oligopeptide (as this formally means many peptides joined together).

Response:

We agree with the reviewer and have made changes accordingly. Please see the revised manuscript.

The sentence suggesting compatibility with solid-phase synthesis just because a Boc group remained intact is pushing credibility. This needs to either proven (by doing the chemistry on a resin) or the claim removed. The use of Pd and other salts is a real issue for SPPS (often blocks the solid-phase), while some linkers are cleaved by cold HFIP. HFIP at 80oC with AgTFA – who knows.

Response:

We agree with the reviewer and have removed the statement regarding the compatibility with SPPS.

By definition (IUPAC) the abbreviation for d-amino acids is a three letter code with no capitals, while the L-amino acids has the first letter capitalised. tLeu needs to be rewritten - italics, superscript.

Response:

We agree with the reviewer and have made changes following the reviewer's instruction.

I would like to see yields given with an $n=3$ i.e. three repeats and +- SD to show robustness of the chemistry. This should be compulsory.

Response:

We have providing the yields obtained from three independent trials. Please see the revised manuscript.

Thank you for editing our manuscript. I hope that you find the revised manuscript suitable for publication in *Nature Communication*. Please do not hesitate to contact me if you have further comments to discuss on any of these points.

Sincerely,

Huan Wang

Encl. (electronically): Revised Manuscript and Supporting Information